# The Prognostic Value of Eight Comorbidity Indices in Older Patients with Cancer: The ELCAPA Cohort Study

**DOI:** 10.3390/cancers14092236

**Published:** 2022-04-29

**Authors:** Florence Canoui-Poitrine, Lauriane Segaux, Marc-Antoine Benderra, Frédégonde About, Christophe Tournigand, Marie Laurent, Philippe Caillet, Etienne Audureau, Emilie Ferrat, Jean-Leon Lagrange, Elena Paillaud, Sylvie Bastuji-Garin

**Affiliations:** 1IMRB, Institut National de la Santé et de la Recherche Médicale (Inserm), Univ Paris Est Creteil, F-94000 Creteil, France; florence.canoui-poitrine@aphp.fr (F.C.-P.); lauriane.segaux@aphp.fr (L.S.); fredegonde.about@aphp.fr (F.A.); christophe.tournigand@aphp.fr (C.T.); marie.laurent@aphp.fr (M.L.); philippe.caillet@aphp.fr (P.C.); etienne.audureau@aphp.fr (E.A.); emilie_frisouille@yahoo.fr (E.F.); lagrange.jeanleon@gmail.com (J.-L.L.); elena.paillaud@aphp.fr (E.P.); sylvie.bastuji-garin@aphp.fr (S.B.-G.); 2Public Health Department & Clinical Research Unit (URC Mondor), Henri-Mondor Hospital, Assistance-Publique Hôpitaux de Paris (AP-HP), F-94010 Creteil, France; 3Department of Medical Oncology, Tenon Hospital, Assistance-Publique Hôpitaux de Paris (AP-HP), F-75020 Paris, France; 4Institut Universitaire de Cancérologie, Sorbonne University, F-75004 Paris, France; 5Department of Medical Oncology, Henri-Mondor Hospital, Assistance-Publique Hôpitaux de Paris (AP-HP), F-94010 Creteil, France; 6Department of Geriatrics, Henri-Mondor/Emile Roux Hospital, Assistance-Publique Hôpitaux de Paris (AP-HP), F-94456 Limeil-Brevannes, France; 7Department of Geriatrics, European Georges Pompidou Hospital, Paris Cancer Institute CARPEM, Assistance-Publique Hôpitaux de Paris (AP-HP), F-75015 Paris, France; 8Department of General Practice, Univ Paris Est Creteil, Université Paris-Est Créteil (UPEC), F-94000 Creteil, France

**Keywords:** comorbidities, indices, mortality, cancer, older patients

## Abstract

**Simple Summary:**

A prognostic assessment is crucial for making cancer treatment decisions. However, the frequent presence of comorbidities makes this assessment particularly challenging in older adults. Many comorbidity indices are available, but none have been developed for older patients with cancer. We assessed the prognostic performance (relative to one-year mortality) of eight comorbidity indices in a cohort of older patients with cancer. All the comorbidity indices were independently associated with 1-year mortality in the whole study population, and had very good discriminant ability. Among patients with metastatic cancer, only CIRS-G was independently associated with 1-year mortality.

**Abstract:**

Background: A prognostic assessment is crucial for making cancer treatment decisions in older patients. We assessed the prognostic performance (relative to one-year mortality) of eight comorbidity indices in a cohort of older patients with cancer. Methods: We studied patients with cancer aged ≥70 included in the Elderly Cancer Patient (ELCAPA) cohort between 2007 and 2010. We assessed seven nonspecific indices (Charlson Comorbidity Index (CCI), three modified versions of the CCI, the Elixhauser Comorbidity Index, the Gagne index, and the Cumulative Illness Rating Scale for Geriatrics (CIRS-G)) and the National Cancer Institute Comorbidity Index. Results: Overall, 510 patients were included. Among patients with nonmetastatic cancer, all the comorbidity indices were independently associated with 1-year mortality (adjusted hazard ratios (aHRs) of 1.44 to 2.51 for one standard deviation increment; *p* < 0.05 for all) and had very good discriminant ability (Harrell’s C > 0.8 for the eight indices), but were poorly calibrated. Among patients with metastatic cancer, only the CIRS-G was independently associated with 1-year mortality (aHR (95% confidence interval): 1.26 [1.06–1.50]). Discriminant ability was moderate (0.61 to 0.70) for the subsets of patients with metastatic cancer and colorectal cancer. Conclusion: Comorbidity indices had strong prognostic value and discriminative ability for one-year mortality in older patients with nonmetastatic cancer, although calibration was poor. In older patients with metastatic cancer, only the CIRS-G was predictive of one-year mortality.

## 1. Introduction

In Europe, the proportion of older adults among patients with newly diagnosed cancer is increasing; in 2020, 60% of these patients were aged 65 or over [1]. Physicians who care for patients with cancer have to make cancer treatment decisions and prognostic assessments, perform follow-ups, and also manage comorbidities. The comorbidity rate among cancer patients increases with age. According to Medicare data, 40% of cancer patients in the USA aged 65 years or older have one comorbidity, and 15% have two or more [2]. Comorbidities are associated with delayed screening for cancer [3,4], the decreased feasibility of cancer treatment [5], and treatment-related complications [5,6,7,8]. To reduce toxicities, new treatment strategies have been developed such as metronomic chemotherapy, which is based on continuous and low doses of chemotherapeutic drugs with dose-dense administration [9]. Many tools for measuring comorbidity burden are available for use in various settings: geriatric inpatients without cancer (e.g., the Charlson Comorbidity Index (CCI) [10]), older community-dwelling patients [11], and nongeriatric patients with cancer (the combined National Cancer Institute (NCI) Comorbidity Index [12]). Moreover, these tools are based on different data sources, including medical records (e.g., the CCI), claims databases (the Elixhauser-van Walraven Comorbidity Index (ECI) [13], and modified versions of the CCI such as the Romano-CCI [13] and the Quan-CCI [14]). In 2012, a comprehensive review identified 21 different comorbidity indices for use with cancer patients and concluded that each had drawbacks in terms of validity, reliability and/or feasibility [15]. None of the indices had been developed specifically for or with older patients with cancer. The Cumulative Illness Rating Scale for Geriatrics (CIRS-G) and the CCI are most frequently used in everyday geriatric oncology: a recent review of the use of geriatric assessment (GA) components in routine oncology practice demonstrated that the CIRS-G and the CCI were applied in 18 (31%) and 20 (34%) of the 58 studies covered, respectively [16]. This use is in line with the recommendation of the International Society of Geriatric Oncology [17].

However, none of these comorbidity indices have been developed in geriatric oncology settings, and so their extension to this patient population is subject to debate. Even though various comorbidity indices [10,13,18,19] are significantly and independently associated with overall mortality [20,21], their validity (in terms of calibration and discriminant ability) has not been explored in geriatric oncology settings. Lastly, comparisons of the various comorbidity indices are scarce, and most that have been compared concern only two indices.

The objective of the present observational study was therefore to compare the ability of eight comorbidity indices to predict one-year mortality in older patients with cancer (overall, by metastatic status, and by tumor site).

## 2. Materials and Methods

The study’s procedures were approved by an investigational review board (*CPP Ile-de-France I*, Paris, France. IRB number: IORG0009918). Informed consent was obtained from all study participants. This observational study is reported in accordance with the “Strengthening the Reporting of Observational Studies in Epidemiology” statement [22].

### 2.1. Design and Setting

The study data came from the Elderly Cancer Patient (ELCAPA) cohort, which has been described in detail elsewhere [23]. Briefly, this prospective, open cohort includes in- and outpatients aged 70 or over who have solid or hematological cancers. The recruitment started at the Henri-Mondor teaching hospital (Créteil, France) in 2007, and was progressively extended to 19 hospitals in the greater Paris area. Patients are referred by oncologists, radiation oncologists, surgeons, or other specialists to the geriatric oncology clinic for a geriatric assessment (GA) prior to a cancer treatment decision. In the present analysis (ELCAPA 04), we included all ELCAPA patients recruited between January 2007 and December 2010. The study inclusion date was taken as the date of the GA.

### 2.2. Data Collection

At baseline, a geriatrician performed a GA; this included a detailed clinical examination, functional tests, and the collection of a blood sample for laboratory tests. Comorbidities were extensively assessed; they included cardiovascular disease (e.g., coronary artery disease, chronic heart failure, and arrhythmia), neuropsychiatric disease (e.g., cerebrovascular disease, dementia, and depression), pulmonary disease, gastrointestinal disease, urinary tract disease, sensory organ impairments, and renal dysfunction. We used these data to compute eight comorbidity indices. Firstly, we computed the original CCI [10] (17 items; range: 0 to 31), the versions of the CCI modified by Romano et al. [13] (15 items; range: 0 to 28) and Quan et al. [24] (12 items; range: 0 to 26), and the CCI adjusted for age [25] (18 items, range 0 to 36). Secondly, we computed the ECI, which was developed initially for claims data [26] and was calibrated by Van Walraven et al. for death in hospital [18] (21 items; range: −19 to 89). Thirdly, we computed the Gagne index [11], which is a combination of the CCI and the ECI (20 items; range: −3 to 25). Fourthly, we computed the CIRS-G, a variant of the CIRS specifically designed for geriatric patients and that rates the severity of any dysfunction across 14 organ systems on a 5-point scale (from 0 (normal) to 4 (very severe dysfunction)) [19] (14 items; range: from 0 to 56). Fifthly, we computed the NCI Comorbidity Index for the breast cancer, prostate cancer and colorectal cancer subgroups. The NCI Comorbidity Index is a cancer-site-specific comorbidity index, in which comorbidities are weighted differently according to the tumor site [12]. The Appendix A shows the eight comorbidity indices with their items and scoring rules.

Demographic characteristics (age and sex), tumor site, metastatic status, and functional status (as assessed by Eastern Cooperative Oncology Group Performance Status (ECOG-PS)) were also systematically recorded in the database.

### 2.3. Main Outcome Measure

The main outcome measure was overall 1-year mortality. Deaths were identified by medical chart review and queries were sent to public record offices.

### 2.4. Statistical Analysis

Patient characteristics were described by the frequency (percentage) of qualitative variables and the mean (standard deviation (SD)) of quantitative variables. We defined complete cases as those having all the data required to calculate the eight comorbidity indices. We used Pearson’s chi-square test or Fisher’s exact test (for qualitative variables) and Student’s *t*-test (for quantitative variables) to compare the characteristics of complete cases with those of cases with missing data.

The dependent variable was death during the first year of follow-up. The analyzed population had all the data required for comparison of all the comorbidity indices. The reference multivariate model was a Cox proportional hazards model adjusted for age, sex, ECOG-PS, and a composite variable including tumor site and metastatic status. The proportional hazard assumption was assessed using Schoenfeld residual plots and tests. Next, an additional (one-by-one) adjustment for each comorbidity index was performed to assess the incremental value in predicting one-year mortality. The comorbidity index data were handled as a continuous variable, the hazard ratio (95% confidence interval (CI)) was calculated for a one SD increment. Discriminant ability (i.e., the model’s ability to distinguish between survivors and nonsurvivors) was assessed using Harrell’s C-index with bootstrapped 95% CIs and two indices developed by Royston and Sauerbrei [27] (the D statistic (95% CI) and R^2^, for which higher values indicate better discriminant ability and which lack censoring bias). The cut-offs for Harrell’s C were defined as follows: 0.50–0.59, poor; 0.60–0.69, moderate; 0.70–0.79, good; 0.80–0.89, very good; and ≥0.90, excellent [28]. There is no consensus cut-off for D or R^2^, although a R^2^ > 0.6 reportedly corresponds to considerable discriminant ability.

We assessed calibration using graphs of individual observed and predicted 1-year survival probabilities; we also assessed whether the slope of the observed probabilities on the predicted event probabilities was equal to 1.

The same methodology was used for assessing the predictive validity of each comorbidity index for the different subgroups (i.e., with and without metastasis, breast, colorectal, and prostate tumor site), which had sufficient sample sizes and events to compute robust multivariate analysis.

### 2.5. Sensitivity Analysis

In order to estimate the missing values for the eight comorbidity indices, we used the multiple multivariate imputation by chained equations procedure in STATA software (v13.0, College Station, TX, USA) with the missing-at-random assumption for age, sex, metastasis status, and tumor site. Twenty imputed datasets were created. Predictive mean matching was used to impute the missing values for quantitative variables.

All tests were two-tailed. The threshold for statistical significance was set to *p* < 0.05. *p* values between 0.05 and 0.10 were considered to be indicative of trends. All statistical analyses were performed using STATA software.

## 3. Results

Of the 644 cancer patients included in the ELCAPA cohort between 2007 and 2010, 134 were excluded because they had missing data required for the calculation of one or more of the indices (Figure 1). A comparison of the included patients (*n* = 510) and excluded patients (*n* = 134) did not reveal any significant differences with regard to age, functional status (assessed on the Activities of Daily Living scale [29]), ECOG-PS, the CIRS-G score, or metastatic status. Compared with the excluded group, the included group had a higher proportion of patients with breast cancer (21.1% versus 9.3%, respectively; *p* = 0.002), a smaller proportion with upper gastrointestinal tract cancer (8.9% versus 15.5%, respectively; *p* = 0.03), and a lower overall 1-year mortality rate (40.5% versus 53.4%, respectively; *p* = 0.003).

Table 1 summarizes the main characteristics of the 510 included patients. In all, 41 items were required to determine the eight indices (as seen in the Appendix A). All scores were correlated with the ECOG-PS (correlation coefficients; 0.30 to 0.62; all *p* < 0.0001) or correlated with each other (from 0.33 (for Quan CCI version × CIRS-G) to 0.9 (for the ECI × the Gagne index); all *p* < 0.001).

### 3.1. Main Analysis

Table 2 and Table 3 show the association of each comorbidity index with one-year mortality and the indices’ calibration and discriminant ability in a multivariate analysis adjusted for age, sex, tumor site, metastatic status, and ECOG-PS for the whole study population and by metastasis status. The proportional hazards assumption was met. In the population as a whole, the CCI and its three derivatives were not independently associated with 1-year overall mortality. Conversely, the ECI, the Gagne index and the CIRS-G were independently associated with 1-year mortality. In the subset of patients with nonmetastatic cancer, all the comorbidity indices were independently associated with 1-year mortality. In the subset of patients with metastatic cancer, only the CIRS-G was associated with 1-year mortality. The slope test indicated a good level of calibration for all comorbidity indices in the study population as a whole and in the subgroup of patients with metastatic cancer. Conversely, the poor calibration for the patients with nonmetastatic cancer indicated that the model’s ability to distinguish between survivors and nonsurvivors did not accurately follow the observed events at each time: models including comorbidity indices overestimated the risk of death for patients with a low comorbidity burden and underestimated the risk of death for patients with a high comorbidity burden (Figure 2). Regarding discriminant ability (i.e., the model’s ability to distinguish between survivors and nonsurvivors, according to Harrell’s C, D, and R^2^), all comorbidity indices performed well in the overall study population and for patients with metastatic cancer, and performed very well for patients with nonmetastatic cancer. However, adding a comorbidity index to the reference model (age, sex, ECOG-PS status, metastatic status and tumor site) led only to a small increase in discriminant ability.

### 3.2. Tumor Site Analysis

All indices were well calibrated in the colorectal and breast cancer subgroups (Table 2 and Table 3). Conversely, all scores had poor calibration in the prostate cancer subgroup. The discriminant ability was excellent for all indices in the breast and prostate subgroups and good in the colorectal subgroup.

### 3.3. Sensitivity Analyses

After the imputation of missing data, the following indices were statistically significant: all the comorbidity indices in the overall population, the ECI, the Gagne index and the CIRS-G in the metastatic cancer group, all the indices in the nonmetastatic cancer group, the Romano CCI version, the ECI, the Gagne index and the CIRS-G in the breast cancer subgroup, and the Gagne index and the CIRS-G in the colorectal and prostate cancer subgroups (Table 4 and Table 5).

## 4. Discussion

Comorbidity indices (namely the original CCI, three modified CCIs, the ECI, the Gagne index, and the CIRS-G) had a strong prognostic value for one-year mortality in older patients with nonmetastatic cancer. However, these indices were poorly calibrated. In metastatic cancer patients, none of the nonspecific comorbidity indices were significantly associated with mortality, with the exception of CIRS-G. By tumor type analysis, only the breast-cancer-specific NCI Comorbidity Index had significant prognostic value, good calibration, and discriminant ability. In the prostate cancer subgroup, only the CIRS-G had strong prognostic and discriminative value, but lacked calibration.

None of the tested comorbidity indices were developed or validated in older patients with cancer. The CCI was developed in a population of 559 inpatients hospitalized for a month at New York Hospital, Cornell Medical Center in 1984 and validated in a cohort of 685 breast cancer patients treated at Yale New Haven Hospital between 1962 and 1969 [10]. The weights of each comorbidity were in accordance with the burden and severity of diseases that were prominent in the 1980s: acquired immunodeficiency syndrome (AIDS), which was a deadly disease in the period, therefore had a high weight. The age-adjusted CCI was developed in a cohort of 225 patients with hypertension or diabetes and who underwent elective surgery between 1982 and 1985 [25]. To apply the CCI to administrative hospital discharge data, Romano et al. [13], developed coding algorithms for the International Classification of Diseases, Ninth Revision. Quan et al. [14] updated the CCI with regard to the population living in the Calgary Health Region (Alberta, Canada) in 2004 (population 1.3 million); the modified CCI has since been validated in six countries [14]. The ECI was developed using medical and administrative data on adult, nonmaternal inpatients hospitalized in 438 acute care hospitals in California in 1992 (*n* = 1,779,167) [26]. Gagne’s index [11] was the only index originally developed in (community-dwelling) older adults; it combines the CCI-Romano and the ECI. Gagne et al.’s development cohort was similar to ours with regard to the mean age (79 and 80, respectively), but only 11% had cancer. The CIRS-G was developed by Linn et al. in 1968 [30] and modified by Miller et al. in 1992 [31] for geropsychiatric practice and research, using a population of 141 elderly outpatients.

Previous studies in various geriatric and cancer settings have found significant independent associations with overall mortality for the CIRS-G, CCI, Romano/CCI, and the ECI [32]. Of the 21 methods for measuring comorbidity in populations with cancer identified in a 2012 review [15], three were found to exhibit good criterion validity, reliability, and feasibility: the CCI, the ECI, and the NCI Comorbidity Index (based on a subset of CCI item plus specific cancer items).

Few studies have evaluated and/or compared several indices simultaneously. In a study of 891 patients that underwent radical cystectomy for bladder cancer, the CCI and the ECI were good predictors of overall 5-year mortality (C-statistic: 0.80 and 0.77, respectively) [33]. A similar comparison in patients with colorectal cancer revealed very good accuracy for the CCI and the ECI (C-statistic: 0.83 and 0.85, respectively) in predicting overall 3-year mortality, taking account of age, sex, and cancer stage [34]. Another analysis of patients with prostate cancer compared five chart-based comorbidity indices (including the CIRS and CCI); each index explained a significant proportion of the variability in overall survival (beyond age), and the CIRS performed slightly better than the CCI [35]. In line with these literature findings, the results of our subgroup analysis by tumor site suggested that all indices (except the CCI in the subgroup of patients with genitourinary tract cancer) were well calibrated. Accuracy was high in the breast and genitourinary cancer subgroups, but lower in the colorectal cancer subgroup. Moreover, the confidence intervals were wide and varied substantially from one tumor site to another. Among patients with metastatic cancer, only the CIRS-G was independently associated with 1-year mortality. This result is difficult to explain. We hypothesize that it can be explained by chance, or by the CIRS-G itself. This index authorizes a comprehensive approach to evaluate comorbidity. It takes into account the severity of the comorbidities, which is important in metastatic situations. Severe or unstable comorbidities may limit the indications of treatment, maybe more than in nonmetastatic situations [32].

### Strengths and Limitations

Our study provided new information by simultaneously comparing eight comorbidity indices in the same population of older patients with cancer. In addition to calculating adjusted hazard ratios, we assessed calibration and discriminant ability. We also performed subgroup analyses. Furthermore, the investigators who determined the indices were blinded to the main outcome measure (1-year mortality), which limited the risk of classification bias.

The limitations of our study included possible selection bias related to patient recruitment at the geriatric oncology clinics. Patients consulting at these clinics might not be representative of the overall population of older patients with cancer. Lastly, our subgroup analyses by tumor site might have lacked power.

## 5. Conclusions

Among older patients with nonmetastatic cancer, all eight tested comorbidity indices were independently associated with 1-year mortality and had very good discriminant ability, but were poorly calibrated. Recalibration is needed before these comorbidity indices can be used as prognostic tools through score revision (changing the item’s weights) or score extension [36].

Among patients with metastatic cancer, only the CIRS-G was associated with 1-year mortality. Its calibration and discriminant ability were good. The CIRS-G may be useful for assessing prognoses related to comorbidities in older patients with metastatic cancer. However, a useful prognostic index must be able to improve patient outcomes and decrease costs and adverse events, relative to the physician’s subjective judgment. Whether or not this outcome is achieved must be determined in randomized, controlled trials.

## Figures and Tables

**Figure 1 cancers-14-02236-f001:**
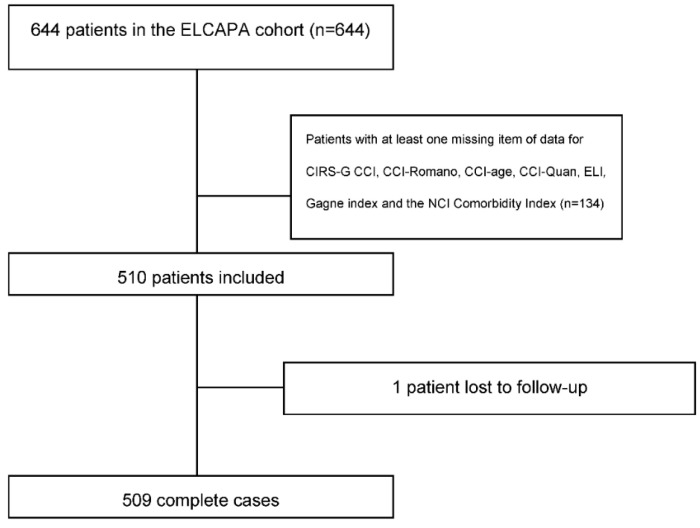
Study flow diagram: 26 missing items of data for the first 644 patients: tumor site (*n* = 9), age (*n* = 1), vital status (*n* = 4), metastasis (*n* = 16). Some patients lacked more than one item of data.

**Figure 2 cancers-14-02236-f002:**
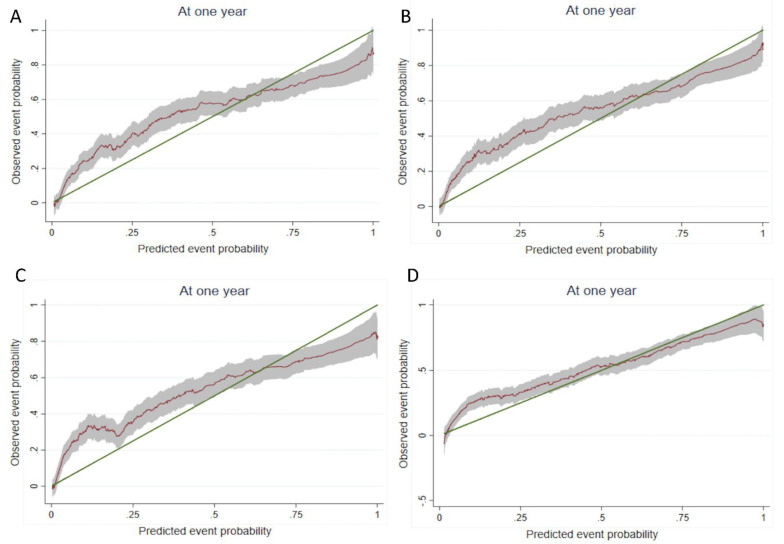
Calibration slopes for (**A**) the original CCI, (**B**) the Gagne index, (**C**) the ECI, and (**D**) the CIRS-G. Smoothed pseudovalues (red line) with 95% CI plotted against predicted event probabilities at one year. The green line is the line of identity, denoting perfect calibration. CCI indicates Charlson Comorbidity Index; ECi indicates Elixhauser-van Walraven Comorbidity Index; CIRG-S indicates Cumulative Illness Rating Scale for Geriatrics.

**Table 1 cancers-14-02236-t001:** Main characteristics of the 510 patients included.

	Patients (*n* = 510)	
	No.	%
Age, median (IQR)	80 (76–84)	
Male sex	233	45.7
ECOG PS ≥ 2	235	50.4
Solid cancer	466	91.6
Metastatic cancer	215	47.3
Haematologic malignancies	43	8.4
Cancer site (*n* = 463)		
-Colorectal	130	25.7
-Breast	107	21.1
-Urinary tract	55	10.9
-Prostate	46	9.1
-Upper digestive tract and liver	45	8.9
-Pancreas	40	7.9
-Others *	40	7.9
Indices		
Charlson Comorbidity Index (CCI), median (IQR)	7 (4–12.5)	
Romano CCI version, median (IQR)	8 (5–13)	
Quan CCI, median (IQR)	5 (3–10)	
Age-adjusted CCI version, median (IQR)	10.5 (8–14)	
Elishauser, median (IQR)	21 (12–32)	
Gagne, median (IQR)	7 (4–10)	
CIRS-G, median (IQR)	12 (8–16)	

* Others: sarcoma (*n* = 9); lung (*n* = 9); skin (*n* = 3); brain (*n* = 2); ovary (*n* = 1); endometrium (*n* = 1); leiomyosarcoma (*n* = 1); mediastina (*n* = 1); larynx (*n* = 1); unknown (*n* = 11); penis (*n* = 1). SD indicates standard deviation; IQR indicates interquartile range; ECOG-PS indicates Eastern Cooperative Oncology Group Performance Status; CIRS-G indicates Cumulative Illness Rating Scale for Geriatrics.

**Table 2 cancers-14-02236-t002:** The association of each comorbidity index with one-year mortality and the indices’ calibration and discriminant ability for the whole study population, metastatic and nonmetastatic cancer.

	Whole Population				Metastatic Cancer				Nonmetastatic Cancer			
	*n* = 510				*n* = 215				*n* = 240			
Models	Hazard Ratio * 95% CI	Calibration	C Harrel	D.Sauerbrei/R^2^	Hazard Ratio * 95% CI	Calibration	C Harrel	D.Sauerbrei/R^2^	Hazard Ratio * 95% CI	Calibration	C Harrel	D.Sauerbrei/R^2^
Initial (Adjusted for age, ECOG-PS and sex)	**-**	0.85	0.7897 (0.77–0.81)	1.89/0.46	-	0.56	0.7242 (0.71–0.77)	1.35/0.30	-	0.49	0.8051 (0.74–0.84)	1.74/0.42
+CCI	1.06 (0.86–1.29)	0.71	0.7893 (0.75–0.81)	1.90/0.46	0.82 (0.64–1.04)	0.31	0.7282 (0.72–0.77)	1.30/0.29	1.93 (1.34–2.77)	0.004	0.8194 (0.76–085)	2.34/0.57
+Romano CCI version	1.13 (0.94–1.36)	0.73	0.7898 (0.76–0.81)	1.92/0.47	0.88 (0.70–1.12)	0.46	0.7257 (0.70–0.78)	1.29/0.28	2.06 (1.44–2.96)	0.001	0.8211 (0.75–0.85)	2.38/0.57
+Quan CCI version	1.06 (0.86–1.31)	0.71	0.7897 (0.76–0.81)	1.91/0.47	0.81 (0.64–1.04)	0.28	0.7281 (0.71–0.76)	1.30/0.29	2.13 (1.48–3.07)	0.002	0.8212 (0.74–0.86)	2.44/0.59
+Age-adjusted CCI	1.08 (0.88–1.33)	0.73	0.7887 (0.75–0.81)	1.93/0.47	0.83 (0.64–1.06)	0.37	0.7287 (0.71–0.79)	1.29/0.28	1.98 (1.37–2.86)	0.002	0.8203 (0.76–0.85)	2.36/0.57
+Elixhauser	1.28 (1.08–1.52)	0.81	0.7938 (0.76–0.81)	2.01/0.49	1.02 (0.80–1.29)	0.80	0.725 (0.70–0.77)	1.38/0.31	2.40 (1.75–3.28)	0.006	0.8377 (0.76–0.87)	2.78/0.65
+Gagne	1.41 (1.19–1.69)	0.74	0.7982 (0.76–0.82)	2.09/0.51	1.12 (0.88–1.41)	0.96	0.7246 (0.70–0.77)	1.37/0.31	2.51 (1.80–3.50)	0.001	0.8419 (0.78–0.88)	2.81/0.65
+CIRS-G	1.29 (1.13–1.48)	0.94	0.8029 (0.76–0.82)	2.01/0.49	1.26 (1.06–1.50)	0.62	0.7402 (0.72–0.80)	1.43/0.33	1.44 (1.02–2.04)	0.18	0.8109 (0.74–0.85)	1.93/0.47

* Adjusted for age, PS status and sex. ECOG-PS indicates Eastern Cooperative Oncology Group Performance Status; CCI indicates Charlson Comorbidity Index; CIRG-S indicates Cumulative Illness Rating Scale for Geriatrics; NCI indicates National Cancer Institute.

**Table 3 cancers-14-02236-t003:** The association of each comorbidity index with one-year mortality and the indices’ calibration and discriminant ability by cancer group.

	Breast				Colorectal				Prostate			
	*n* = 106				*n* = 130				*n* = 46			
Models	Hazard Ratio * 95% CI	Calibration	C Harrel	D.Sauerbrei/R^2^	Hazard Ratio * 95% CI	Calibration	C Harrel	D.Sauerbrei/R^2^	Hazard Ratio * 95% CI	Calibration	C Harrel	D.Sauerbrei/R^2^
Initial (Adjusted for age, ECOG-PS and sex)	-	0.85	0.8683 (0.79–0.91)	2.57/0.61	-	0.71	0.7505 (0.66–0.81)	1.52/0.36	-	<0.001	0.8950 (0.77–0.93)	2.82/0.66
+CCI	0.83 (0.33–2.09)	0.93	0.8683 (0.78–0.90)	2.74/0.64	0.78 (0.56–1.10)	0.96	0.7425 (0.65–0.80)	1.72/0.41	1.16 (0.43–3.13)	<0.001	0.9037 (0.87–0.94)	3.03/0.69
+Romano CCI version	1.26 (0.58–2.76)	0.49	0.8672 (0.78–0.90)	2.50/0.60	0.84 (0.59–1.20)	0.89	0.7456 (0.67–0.81)	1.77/0.43	0.91 (0.34–2.45)	<0.001	0.9081 (0.82–0.94)	3.07/0.69
+Quan CCI version	1.13 (0.45–2.83)	0.74	0.8680 (0.79–0.90)	2.52/0.60	0.83 (0.58–1.21)	0.85	0.7461 (0.67–0.81)	1.73/0.42	0.52 (0.11–2.45)	<0.001	0.9212 (0.82–0.95)	3.78/0.77
+Age-adjusted CCI	0.86 (0.33–2.21)	0.92	0.8705 (0.79–0.91)	2.74/0.64	0.78 (0.55–1.12)	0.96	0.7436 (0.66–0.80)	1.74/0.42	1.26 (0.45–3.49)	<0.001	0.9037 (0.79–0.93)	3.16/0.70
+Elixhauser	1.58 (0.83–2.98)	0.01	0.8718 (0.80–0.91)	2.50/0.60	1.07 (0.73–1.56)	0.86	0.7522 (0.67–0.81)	1.46/0.34	0.72 (0.30–1.74)	<0.001	0.9125 (0.82–0.95)	3.68/0.76
+Gagne	1.41 (0.70–2.83)	0.24	0.8675 (0.80–0.90)	2.47/0.59	1.06 (0.74–1.54)	0.82	0.7547 (0.67–0.81)	4.52/0.36	0.88 (0.38–2.06)	<0.001	0.8993 (0.81–0.93)	2.91/0.67
+CIRS-G	1.38 (0.79–2.41)	0.002	0.8772 (0.78–0.92)	2.84/0.66	0.98 (0.73–1.33)	0.58	0.7510 (0.66–0.81)	1.54/0.36	2.80 (1.05–7.49)	<0.001	0.9365 (0.88–0.98)	4.55/0.83
+NCI breast	1.02 (1.00–1.04)	0.36	0.8691 (0.76–0.91)	2.49/0.60								
+NCI Colorectal					1.04 (0.97–1.11)	0.99	0.7569(0.68–0.82)	1.39/0.32				
+NCI prostate									1.06 (0.88–1.27)	<0.001	0.8993 (0.81–0.93)	3.09/0.70

* Adjusted for age, PS status and sex. ECOG-PS indicates Eastern Cooperative Oncology Group Performance Status; CCI indicates Charlson Comorbidity Index; CIRG-S indicates Cumulative Illness Rating Scale for Geriatrics; NCI indicates National Cancer Institute.

**Table 4 cancers-14-02236-t004:** The association of each comorbidity index with one-year mortality for the whole study population, metastatic and nonmetastatic cancer after the imputation of missing data.

	Start Population				Metastatic Cancer				Nonmetastatic Cancer			
	*n* = 618				*n* = 269				*n* = 292			
Models	Hazard Ratio * 95% CI	Calibra tion	C Harrel	D.Sauerbrei/R^2^	Hazard Ratio * 95% CI	Calibra tion	C Harrel	D.Sauerbrei/R^2^	Hazard Ratio * 95% CI	Calibration	C Harrel	D.Sauerbrei/R^2^
Initial (Adjusted for age, ECOG-PS and sex)	**-**				**-**				**-**			
+CCI	1.27(1.05–1.52)				1.05(0.86–1.29)				1.91(1.29–2.81)			
+Romano CCI version	1.35(1.13–1.60)				1.12(0.92–1.37)				2.06(1.42–3.00)			
+Quan CCI version	1.33(1.11–1.61)				1.09(0.89–1.34)				2.16(1.47–3.18)			
+Age-adjusted CCI	1.29(1.07–1.55)				1.07(0.86–1.32)				1.96(1.32–2.91)			
+Elixhauser	1.49(1.27–1.76)				1.23(1.01–1.50)				2.40(1.75–3.28)			
+Gagne	1.69(1.43–2.00)				1.41(1.15–1.72)				2.55(1.82–3.58)			
+CIRS-G	1.52(1.37–1.70)				1.56(1.36–1.79)				1.57(1.21–2.04)			

* Adjusted for age, PS status and sex. ECOG-PS indicates Eastern Cooperative Oncology Group Performance Status; CCI indicates Charlson Comorbidity Index; CIRG-S indicates Cumulative Illness Rating Scale for Geriatrics; NCI indicates National Cancer Institute.

**Table 5 cancers-14-02236-t005:** The association of each comorbidity index with one-year mortality by cancer group after the imputation of missing data.

	Breast				Colorectal				Prostate			
	*n* = 115				*n* = 130				*n* = 54			
Models	Hazard Ratio * 95% CI	Calibra tion	C Harrel	D.Sauerbrei/R^2^	Hazard Ratio * 95% CI	Calibra tion	C Harrel	D.Sauerbrei/R^2^	Hazard Ratio * 95% CI	Calibra tion	C Harrel	D.Sauerbrei/R^2^
Initial (Adjusted for age, ECOG-PS and sex)	**-**				**-**				**-**			
+CCI	1.63(0.82–3.25)				1.06(0.76–1.49)				1.40(0.67–2.89)			
+Romano CCI version	1.99(1.04–3.82)				1.19(0.85–1.66)				1.55(0.73–3.33)			
+Quan CCI version	2.00(0.91–4.40)				1.25(0.87–1.79)				1.28(0.51–3.24)			
+Age-adjusted CCI	1.71(0.85–3.41)				1.07(0.76–1.52)				1.42(0.68–2.97)			
+Elixhauser	1.90(1.10–3.27)				1.35(0.97–1.88)				1.48(0.83–2.64)			
+Gagne	1.90(1.06–3.41)				1.52(1.07–2.16)				1.92(1.07–3.45)			
+CIRS-G	1.90(1.17–3.11)				1.45(1.12–1.87)				2.56(1.31–4.98)			

* Adjusted for age, PS status and sex. ECOG-PS indicates Eastern Cooperative Oncology Group Performance Status; CCI indicates Charlson Comorbidity Index; CIRG-S indicates Cumulative Illness Rating Scale for Geriatrics; NCI indicates National Cancer Institute.

## Data Availability

Restrictions apply to the availability of these data. Data were obtained from the ELCAPA Study Group and are available from the corresponding author with the permission of the ELCAPA Study Group investigators.

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
