# Peer review of "The Prognostic Value of Eight Comorbidity Indices in Older Patients with Cancer: The ELCAPA Cohort Study"

_cancers, 2022, doi:10.3390/cancers14092236_

Round 1
Reviewer 1 Report
In this study, the authors assessed the prognostic performance (relative to one-year mortality) of eight comorbidity indices in a cohort of older patients with cancer. They studied 510 patients with cancer aged ≥70 included in the ELderly CAncer PAtient (ELCAPA) cohort and assessed seven nonspecific indices (Charlson Comorbidity Index (CCI), three modified versions of CCI, the Elixhauser Comorbidity Index, the Gagne index, and the Cumulative Illness Rating Scale for Geriatrics (CIRS-G)) and the National Cancer Institute Comorbidity Index.
Among patients with nonmetastatic cancer, all the comorbidity indices were independently associated with 1-year mortality and had very good discriminant ability (Harrell’s C >0.8 for the eight indices) but were poorly calibrated. Among patients with metastatic cancer, only the CIRS-G was independently associated with 1-year mortality (aHR [95% confidence interval]: 1.26 [1.06-1.50]). Discriminant ability was moderate (0.61 to 0.70) for the subsets of patients with metastatic cancer and colorectal cancer. They concluded that comorbidity indices had strong prognostic value and the discriminative ability for one-year mortality in older patients with nonmetastatic cancer. In older patients with metastatic cancer, only the CIRS-G is predictive of one-year mortality.
The study is of interest and of clinical impact.
I have only a comment regarding the older patients with cancer. In the last years the concept of "metronomic chemotherapy", which is based on the chronic administration of chemotherapeutic agents at low doses without prolonged drug-free breaks to optimize the antitumor properties of the drug and reduce toxicities, has been introduced into oncology. It has been speculated that metronomic chemotherapy could be of major impact on older patients with cancer since older patients might be a fragile population. The authors should recall literature data on the safety and efficacy of published metronomic schedule such as metronomic capecitabine in patients with hepatocellular carcinoma as previously reported (Metronomic capecitabine as second-line treatment in hepatocellular carcinoma after sorafenib failure. Dig Liver Dis. 2015 Jun;47(6):518-22).
Author Response
Dear reviewer,
Thank you for your comments. We have added in the introduction a part on metronomic chemotherapy and we recalled literature data with a new reference
(ref : Cazzaniga ME, Cordani N, Capici S, Cogliati V, Riva F, Cerrito MG. Metronomic Chemotherapy. Cancers (Basel). 6 mai 2021;13(9):2236)
Reviewer 2 Report
With great interest I read the manuscript of the prognostic value of eight comorbidity indices in older patients with cancer: the ELCAPA cohort study. My compliments for this important analysis. It is well written and although a complex statistical analysis clear explained. The results are important for progression for medical decision in older patients with cancer.
I would recommend in the discussion part add when is mentioned "poor calibration" if this means an overestimating or underestimation of the 1 year mortality.
Can you add how recalibration can be achieved?
How relevant is to use the original CCI? For example: AIDS which was a deadly disease in the period it CCI was developed. For the aim of the study probably it is more useful only use the comorbidity indices relevant for clinical practice at this moment.
Author Response
Dear reviewer,
Thank you for your comments and your suggestions.
- We added comments on poor calibration : "models including comorbidity indices overestimated the risk of death for patients with a low comorbidity burden and underestimated the risk of death for patients with a high comorbidity burden"
- for recalibration, we added that we suggest "score revision (change of item’s weights) or score extension"
- finally, in the discussion we added that the weights of each comorbidities were in accordance with the burden and severity of each disease in the 80’s (AIDS)
Please find attached the manuscript revised, taking into account your comments